# Senescence-Independent Anti-Inflammatory Activity of the Senolytic Drugs Dasatinib, Navitoclax, and Venetoclax in Zebrafish Models of Chronic Inflammation

**DOI:** 10.3390/ijms231810468

**Published:** 2022-09-09

**Authors:** David Hernández-Silva, Joaquín Cantón-Sandoval, Francisco Juan Martínez-Navarro, Horacio Pérez-Sánchez, Sofia de Oliveira, Victoriano Mulero, Francisca Alcaraz-Pérez, María Luisa Cayuela

**Affiliations:** 1Telomerase Cancer and Aging Group (TCAG), Hospital Clínico Universitario Virgen de la Arrixaca, 30120 Murcia, Spain; 2Instituto Murciano de Investigación Biosanitaria-Arrixaca (IMIB-Arrixaca), 30120 Murcia, Spain; 3Structural Bioinformatics and High-Performance Computing Research Group (BIO-HPC), Computer Engineering Department, Universidad Católica de Murcia (UCAM), Guadalupe, 30107 Murcia, Spain; 4Departamento de Biología Celular e Histología, Facultad de Biología, Universidad de Murcia, 30100 Murcia, Spain; 5Centro de Investigación Biomédica en Red de Enfermedades Raras (CIBERER), ISCIII, 30100 Murcia, Spain; 6Department of Developmental and Molecular Biology, Albert Einstein College of Medicine, Bronx, NY 10461, USA; 7Department of Medicine (Hepatology), Albert Einstein College of Medicine, Bronx, NY 10461, USA

**Keywords:** aging, senolytics, chronic inflammation, metainflammation, zebrafish

## Abstract

Telomere shortening is the main molecular mechanism of aging, but not the only one. The adaptive immune system also ages, and older organisms tend to develop a chronic pro-inflammatory status with low-grade inflammation characterized by chronic activation of the innate immune system, called inflammaging. One of the main stimuli that fuels inflammaging is a high nutrient intake, triggering a metabolic inflammation process called metainflammation. In this study, we report the anti-inflammatory activity of several senolytic drugs in the context of chronic inflammation, by using two different zebrafish models: (i) a chronic skin inflammation model with a hypomorphic mutation in *spint1a*, the gene encoding the serine protease inhibitor, kunitz-type, 1a (also known as *hai1a*) and (ii) a non-alcoholic fatty liver disease/non-alcoholic steatohepatitis (NAFLD/NASH) model with inflammation induced by a high-fat diet. Our results show that, although these models do not manifest premature aging, the senolytic drugs dasatinib, navitoclax, and venetoclax have an anti-inflammatory effect that results in the amelioration of chronic inflammation.

## 1. Introduction

Life expectancy has doubled over the last century, increasing the percentage of older people in the general population, so understanding aging has become a public health priority [1]. Aging is a complex biological process manifested by the deterioration of a series of physiological systems associated with an increased predisposition to many diseases [2]. The main molecular mechanism of aging is telomere shortening, which results in chromosome instability, replicative senescence, and/or apoptosis [3,4]. In the adaptive immune system, age-associated telomere shortening leads to the state of immunosenescence, with cellular senescence that culminates in cessation of replication [5]. This phenomenon is associated with lymphopenia, the progressive depletion of naïve T cells and the reduced proliferation ability of T cells [6,7]. Despite immunosenescence, it has been observed that older organisms tend to develop a chronic pro-inflammatory status with low-grade inflammation, called inflammaging, characterized by high levels of pro-inflammatory markers in cells and tissues, and chronic activation of the innate immune system, even in the absence of risk factors and clinically active diseases [8]. One of the main stimuli that fuels inflammaging through the so-called metainflammation process is nutrient excess [9]. High nutrient intake is a critical contributor to the onset of insulin resistance and, specifically, excessive pro-inflammatory fatty acids result in an increased activation of inflammatory responses that affect adipose tissue, liver, pancreas, muscle, and brain [10].

Having a longer life expectancy but suffering from age-related diseases is not desirable. Geroscience is a new research field focused on understanding the basic mechanisms driving aging and the link between aging and age-related chronic diseases. Its ultimate goal is to provide novel preventive or diagnostic measures that can reduce the burden of age-related disease and disability to increase the healthy life expectancy [11]. Both immunosenescence and inflammaging result in a diminished capacity of the immune system to remove senescent cells [12,13]. This leads to an accumulation of senescent cells in tissues that contributes to the development of many aging-related diseases [14,15]. Several approaches have been explored to fight diseases associated with cellular senescence, including fasting, exercise, and senolytic drugs to selectively induce apoptosis in senescent cells [16].

The zebrafish model presents many advantages, including the ability to easily maintain large stocks of fish, fast embryonic development ex utero, high transparency at the early developmental stages, and conserved signaling pathways. All these characteristics make it an excellent model to study several human diseases [17,18,19]. The aim of this study is to evaluate the anti-aging effects of three senolytic drugs (Dasatinib, Navitoclax, and Venetoclax) in a context of chronic inflammation to prevent the associated premature aging, by using zebrafish models of inflammaging and metainflammation. The first one is a chronic skin inflammation zebrafish model with a hypomorphic mutation of *spint1a*, the gene encoding the serine protease inhibitor, kunitz-type, 1a (also known as *hai1a*). Spint1a-deficient larvae show neutrophil infiltration in the skin, keratinocyte hyperproliferation that results in aggregate foci, epithelial integrity disruption, and skin inflammation [20,21]. This inflammatory process is mediated by parthanatos cell death as a consequence of hyperactivation of poly (ADP-ribose) polymerase 1 (Parp1) in response to ROS-induced DNA damage and fueled by nicotinamide phosphoribosyltransferase (NAMPT)-derived NAD^+^ [22].

The second model used is a non-alcoholic fatty liver disease/non-alcoholic steatohepatitis (NAFLD/NASH) zebrafish model with metainflammation induced by a high-fat diet. Fish fed a high-fat diet develop non-resolving inflammation in the liver with increased neutrophil infiltration, altered macrophage polarization, reduced T-cell density, and enhanced cancer progression [23]. Our results show that, although these models do not yet manifest premature aging, senolytics have an anti-inflammatory effect that results in the improvement of the phenotype.

## 2. Results

### 2.1. Dasatinib, Navitoclax, and Venetoclax Ameliorate Skin Inflammation in Spint1a-Deficient Larvae

Although it is well-established that chronic inflammation leads to premature aging, we firstly checked the aging stage in the Spint1a-deficient model at 3 days post-fertilization (dpf). Despite the fact that Spint1a-deficient larvae showed high levels of cell death in the skin (Figure 1A), telomere length was not affected (Figure 1B). Consequently, the senescence-associated b-galactosidase (SA b-gal) staining revealed development-associated senescence in the head and muscle of both wild type and Spint1a-deficient larvae (Appendix A), whereas no staining was observed in the skin (Figure 1C). Therefore, the Spint1a-deficient zebrafish line does not show premature aging.

Surprisingly, however, treatment of larvae with the senolytic drugs dasatinib, navitoclax, and venetoclax for 48 h (Figure 2A) was able to decrease neutrophil dispersion, assayed in a transgenic line with labelled neutrophils (lyz:DsRED2) (Figure 2B), as well as the number of keratinocyte aggregates (Figure 2C), although dasatinib was less potent than navitoclax and venetoclax. These results were further supported by the robust reduction of the mRNA levels of the genes encoding pro-inflammatory interleukin-1β (Il1b), tumor necrosis factor α (Tnfa), and chemokine (C-X-C motif) ligand 8b, duplicate 1 (Cxcl8b.1) by all senolytics, except for tnfa by navitoclax (Figure 2D). These results point to a new anti-inflammatory potential of dasatinib, navitoclax, and venetoclax in the context of chronic skin inflammation.

### 2.2. Navitoclax Dampens Emergency Myelopoiesis in a Zebrafish Model of NAFLD/NASH

To confirm the anti-inflammatory effect of senolytics, we also checked the effect of navitoclax in another model of inflammation, the NAFLD/NASH zebrafish larvae model. This is a model with metainflammation induced by a high-cholesterol diet (HCD). We first evaluated aging by quantifying the telomere length in 16-dpf larvae fed with an HCD for 11 days, and the result did not show telomere shortening in larvae fed with the HCD (Figure 3A). The treatment of the HCD-fed group with navitoclax for 4 days (Figure 3B) reversed neutrophilia and monocytosis in the HCD-fed larvae (Figure 3C). Unexpectedly, navitoclax failed to dampen the transcript levels of il1b, tnfa, and cxcl8b.1 genes (Figure 3D). These results confirm the senescence-independent therapeutic benefits of navitoclax in the context of chronic inflammation.

## 3. Discussion

Understanding the basic mechanisms driving aging and the link between aging and age-related chronic diseases is a necessary requirement to increase healthy life expectancy [11]. The main mechanism responsible for aging is telomere shortening through replicative senescence [3,4]. The immune system, which demands a high rate of cell proliferation and renewal, is one of the most affected by telomere shortening and also ages in a process known as immunosenescence [5,6,7]. In addition, a chronic pro-inflammatory status with low-grade inflammation and chronic activation of the innate immune system has been observed in older organisms. This process is called inflammaging [8], and has been recognized as another molecular mechanism that influences aging. Both immunosenescence and inflammaging result in a diminished capacity of the immune system to remove senescent cells [12,13], leading to an accumulation in tissues that contributes to the development of many aging-related diseases [14,15]. Due to its characteristics, the zebrafish is an excellent model for the study of aging, as the age-associated molecular mechanisms are highly conserved [24]. As in aged humans [25], as telomeres shorten, the number of senescent cells increases in some tissues [26,27]. Recently, it has been reported that the recombination activating gene 1 (*rag1*) mutant line is an excellent model of immunosenescence [28]. Compared to their wild-type siblings, the Rag1-deficient zebrafish show (i) a premature aging phenotype, with a reduced lifespan; (ii) a higher incidence of cell cycle arrest and apoptosis; (iii) a greater amount of phosphorylated histone H2AX and oxidative stress; (iv) an upregulated expression of senescence-related genes and senescence-associated b-galactosidase (SA b-gal) activity; (v) diminished telomere length; and (vi) abnormal self-renewal and repair capacities in the retina and liver. As expected, the treatment of *rag1*^−−^ fish with the senolytic navitoclax can reduce both the expression of senescence markers and SA b-gal staining in the skin [29].

Studying aging is a lengthy process, so obtaining models of premature aging at short ages would be of great interest. The Spint1a-deficient model shows neutrophil infiltration in the skin, keratinocyte aggregate foci, epithelial integrity disruption, and skin inflammation [20,21] through parthanatos cell death as a consequence of hyperactivation of PARP1 in response to ROS-induced DNA damage [22]. Although this state of chronic skin inflammation was not sufficient to induce premature aging in larvae, as they did not exhibit telomere shortening or increased cellular senescence (Figure 1), the model did allow us to evaluate the effect of senolytics in the context of chronic inflammation. Accordingly, two days of treatment with the senolytics dasatinib, navitoclax, and venetoclax was sufficient to ameliorate the phenotype of skin inflammation in terms of neutrophil skin infiltration, keratinocyte aggregate foci, and the mRNA levels of genes encoding major pro-inflammatory cytokines (Figure 2). In view of these surprising results, we checked the effect of navitoclax in a context of metabolic inflammation, as it is known that nutrient excess is the main stimulus that triggers inflammaging [9,10]. We used a zebrafish larvae model of metainflammation induced by a high-cholesterol diet (HCD), previously described as the NAFLD/NASH zebrafish model, where the HCD promotes non-resolving inflammation in the liver and enhances cancer progression [23].

As in the previous model, the quantification of telomere length did not show significant changes. However, navitoclax was again able to ameliorate neutrophilia and monocytosis, but it failed to dampen the transcript levels of genes encoding pro-inflammatory mediators. These results are surprising and suggest, on the one hand, that the anti-inflammatory activity of navitoclax may depend on the inflammatory insult and, on the other hand, that it might directly ameliorate emergency myelopoiesis. These results are not fully unexpected, since navitoclax has been shown to effectively remove senescent bone marrow hematopoietic stem cells (HSCs) in aged mice, mitigating premature aging of the hematopoietic system [30]. Therefore, our results pave the way for future studies aimed at investigating whether emergency hematopoiesis promotes the senescence of HSCs or navitoclax is able to regulate hematopoiesis by a senolytic-independent mechanism. 

To date, dasatinib, navitoclax, and venetoclax are well documented as drugs with senolytic activity, inducing apoptosis, specifically in senescent cells [31,32,33]. Although their senolytic activity usually ameliorates inflammation by reducing the senescence-associated secretory phenotype (SASP), which is associated with inflammation, senolytic-independent anti-inflammatory activities have not been reported to date. However, it is not uncommon for a molecule to have different properties. In fact, the senolytic potential of curcumin on senescent intervertebral disc cells by reducing the SASP and back pain by modulating the Nrf2 and NFkB pathways has been recently reported [34,35]. In addition, curcumin has also been demonstrated to be effective as an adjuvant therapy for psoriasis, reducing skin inflammation and serum IL-22 levels [36], these effects being independent of its senolytic activity. Something similar occurs with quercetin, which has been shown to reduce the inflammatory response in patients undergoing coronary artery bypass surgery following an acute coronary syndrome, and also improves endothelial function by eliminating senescent vascular endothelial cells [37]. The multipurpose properties of these nutraceuticals support our results, which point to an anti-inflammatory effect of dasatinib, navitoclax, and venetoclax, acting even before the cell surrounded by a chronic inflammatory microenvironment becomes senescent. Although the mechanism for this new anti-inflammatory effect is not yet sufficiently detailed and more experiments are needed, these drugs could be repurposed as anti-inflammatory agents to treat chronic inflammatory disorders.

## 4. Materials and Methods

### 4.1. Maintenance of Zebrafi

Details of husbandry and environmental conditions are available on protocols io (dx.doi.org/10.17504/protocols.io.br4mm8u6, (accessed on 4 February 2021)). The transgenic zebrafish lines Tg(lyz:dsRED2)nz50,5 [38] and Tg(mpeg1:EGFP)gl22 [39] have been described previously. The mutant zebrafish line spint1ahi2217Tg/hi2217Tg [20] was isolated from an insertional mutagenesis screen.

### 4.2. Diet Preparation and Feeding of Zebrafish Larvae

Prior to any experimental procedure, the larvae were fasted for 24 h. At 3 dpf, the larvae were pre-selected for mpeg1:EGFP (green macrophages), and lyz:dsRED2 (red neutrophils), using a fluorescence magnifier (Leica M205 FCA) equipped with a digital camera (Leica DFC 365 FX), and green and red fluorescence filters. Normal and high-cholesterol diets (ND and HCD, respectively) were prepared as described previously [40], using Golden Pearl Diet 5–50 nm—Active Spheres. At 5 dpf, zebrafish larvae were separated in different tanks and maintained in the system. They were kept in medium size tanks with a density of 80 larvae per tank and fed for 8 days with ND or HCD (4 mg per day). At 13 dpf, the larvae were separated into small breeding boxes (20–30 larvae) and treated with vehicle (0.1% DMSO) or navitoclax (10 µM in 0.1% DMSO) for 3 days, with daily water renewal, and fed with ND or HCD (2.5 mg per day). 

### 4.3. Senolytic Treatment

The senolytics navitoclax, dasatinib, and venetoclax were re-suspended to 1mM in DMSO and added to 24 hpf *spint1a*^+/+^ or *spint1a*^−/−^ larvae, or to 13 dpf larvae fed with a HCD. The larvae were treated for 3 days with the different treatments, navitoclax (10 μM in 0.1% DMSO), dasatinib (1 μM in 0.1% DMSO), venetoclax (10 μM in 0.1% DMSO), or vehicle alone (0.1% DMSO), and the water was renewed daily.

### 4.4. Imaging of Zebrafish Larvae

Live imaging of 3 or 16 dpf larvae was obtained employing buffered tricaine (200 μg/mL) dissolved in egg water. Images of 3 dpf larvae were captured with a fluorescence magnifying glass (Leica M205 FCA) equipped with a digital camera (Leica DFC 365 FX) and set up with green and red fluorescent filters. The images were analyzed to quantify the number of neutrophils, their distribution in the larvae, the number of macrophages, and the keratinocyte aggregates in a common region of interest (ROI) indicated in each figure. For 16 dpf larvae, images were taken by using a Nikon CSU-W1 spinning disk confocal and analyzed with Imaris software (version 9.7.2).

### 4.5. Analysis of Gene Expression

Total RNA was extracted from 3 or 16 dpf larvae with TRIzol reagent (Thermo Fisher Scientific, Waltham, MA, USA), using the Direct-zol RNA Miniprep Kits (zymo research R2050) following the manufacturer’s instructions. The SuperScript™ VILO™ cDNA Synthesis Kit (Invitrogen, Waltham, MA, USA) was used to synthesize first-strand cDNA following the manufacturer’s instructions. Real-time PCR was performed as previously described [29]. The primers used are shown in the Appendix A. In all cases, each PCR was performed with triplicate samples and repeated with at least two independent samples.

### 4.6. Senescence Associated B-Galactosidase (SA B-Gal) Activity Assay

At 24 hpf, larvae were treated with 1-phenyl 2-thiourea (PTU) (3.8 μM) to avoid the formation of melanocytes. After 3 days, the larvae were fixed in 4% paraformaldehyde (PFA) at 4 °C overnight with stirring, then washed 3 times with PBS (pH 7.4) and a fourth wash with PBS (pH 6.0) at 4 °C. Staining was carried out overnight at 37 °C with staining solution (40 mM phosphate/citrate buffer (pH 6.0), 150 mM NaCl, 5 mM potassium ferrocyanide, 5 mM potassium ferricyanide, MgCl_2_ 2 mM and 1 mg/mL X-gal in DMSO) and stirring. After staining, the larvae were transferred to a Petri dish with PBS (pH 7.0) and imaged as detailed above. The quantification of SA b-gal activity staining was performed with ImageJ software by calculating the percentage of blue area at the indicated ROI.

### 4.7. Acridine Orange (AO) Staining Assay

The cell death level was measured by acridine orange (AO) staining as described in the ZFIN Protocol Wiki (https://zfin.atlassian.net/wiki/spaces/prot/overview (accessed on 12 November 2009)). Zebrafish larvae at 3 dpf were incubated in E3 medium with Acridine Orange (Sigma A6014) 10 mg/mL for 30 min at room temperature (RT) The staining was stopped by washing 3 times for 10 min. After staining, the larvae were transferred to a Petri dish with E3 and imaged as detailed above. 

### 4.8. Telomere Length Measurement

gDNA was obtained by using the Wizard Genomic DNA Purification Kit (Promega, Madison, WI, USA) following the manufacturer’s instructions. Telomere length was measured by qPCR following the protocol described by Lau [41], using the primers described in the Appendix A, and 40 ng of gDNA as template. In all cases, each PCR was performed at least twice, with triplicate samples.

### 4.9. Statistical Analysis

Statistical analysis was performed by using GraphPad Prism 8. The differences between the two samples were analyzed by parametric or non-parametric test. Data with more than two samples were analyzed by analysis of variance (ANOVA) or mixed-effect analysis (see Figure legends for further details). 

## Figures and Tables

**Figure 1 ijms-23-10468-f001:**
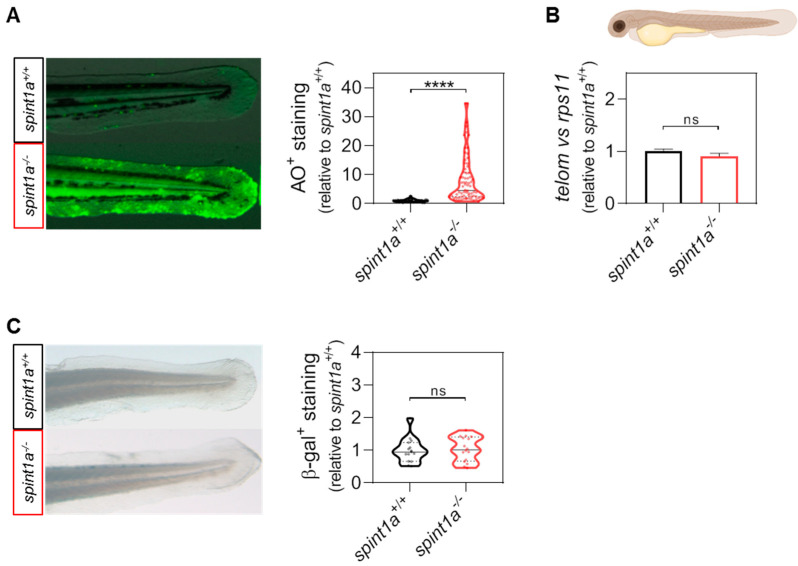
Spint1a-deficient zebrafish larvae do not show premature aging at 3 dpf. (**A**) Representative images and quantification of the cell death levels by acridine orange (AO) staining assay in the tail of wild type (wt) and Spint1a-deficient larvae. The violin plots with the median shown as a horizontal line show the distribution of AO^+^ staining and are overlaid with the raw data, where each dot represents an individual. (**B**) The telomere length was measured in whole zebrafish larvae by qPCR using 40 ng of gDNA and determined as the telomere content relative to the single copy gene rps11. The graph shows the mean ± SEM of 25 pooled larvae (*n* = 25) and triplicate samples from 2 independent experiments (*n* = 2). (**C**) Quantification of the cellular senescence levels by senescence-associated b-galactosidase (SA b-gal) staining assay in the tail of wt and Spint1a-deficient larvae. The violin plots with the median shown as a horizontal line show the distribution of b-gal^+^ staining and are overlaid with the raw data, where each dot represents an individual. ns, non-significant; **** *p* < 0.0001, according to unpaired *t*-test with Welch’s correction (**A**) and unpaired *t*-test (**B**,**C**). Scale bars, 400 μm.

**Figure 2 ijms-23-10468-f002:**
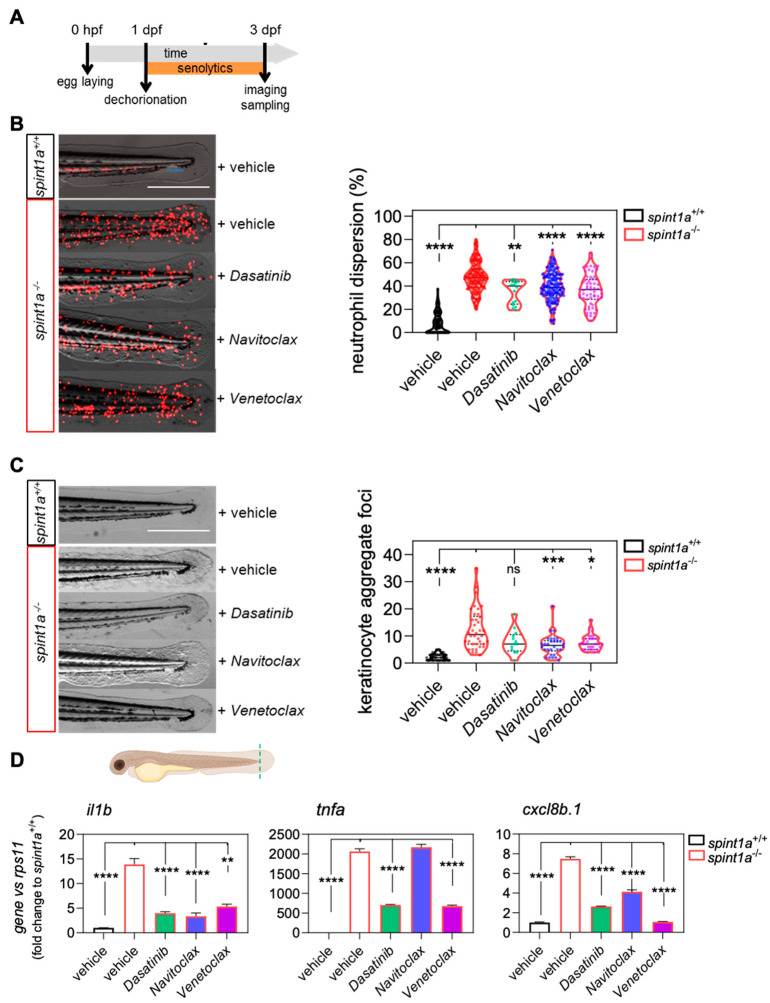
Senolytics ameliorate the skin inflammation phenotype of Spint1a-deficient larvae. (**A**) General workflow for the treatment with senolytics of the chronic skin inflammation zebrafish model (spint1a^−/−^). (**B**) Representative images of neutrophil phenotype and quantification of the number of neutrophils out of the CHT of 3-dpf larvae with neutrophils labeled in red (lyz:DsRED2). (**C**) Representative images of skin phenotype and quantification of the keratinocyte aggregate foci at the skin. The violin plots with the median shown as a horizontal line show the distribution of DsRED2^+^ cells and keratinocyte aggregate foci, and are overlaid with the raw data, where each dot represents an individual. (**D**) The mRNA level of il1b, tnfa, cxcl8b.1 were determined by RT-qPCR and normalized against rps11 in fin folds (indicated at the scheme). The bars show the mean ± SEM of 20 pooled larvae (*n* = 20) and triplicate samples from 2 independent experiments (*n* = 2). ns, non-significant; * *p* < 0.05; ** *p* < 0.01; *** *p* < 0.001; **** *p* < 0.0001 according to Kruskal-Wallis followed by Dunn’s multiple comparison test. CHT: caudal hematopoietic tissue. Scale bars, 400 μm.

**Figure 3 ijms-23-10468-f003:**
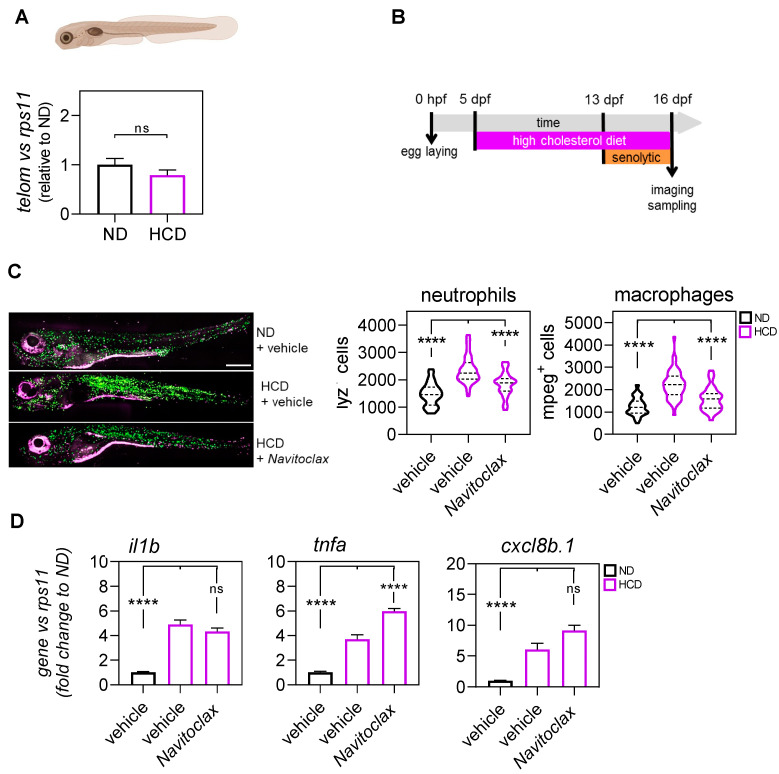
Evaluation of the anti-inflammation effect of navitoclax in the high-cholesterol diet (HCD)-induced metainflammation zebrafish model. (**A**) The telomere length was measured in 16 dpf zebrafish larvae by qPCR using 40 ng of gDNA and determined as the telomere content relative to the single copy gene rps11. The graph shows the mean ± SEM of 25 pooled larvae (*n* = 25) and triplicate samples from 3 independent experiments (*n* = 3). ND (normal diet). HCD (high-cholesterol diet). (**B**) General workflow for the treatment with navitoclax of the zebrafish model with metainflammation induced by HCD. (**C**) Representative images and quantification of the total number of neutrophils (purple) and macrophages (green) of 16 dpf (lyz:DsRED2; mpeg1:EGFP) larvae after 3 days of treatment with navitoclax. The violin plots with the median shown as a horizontal line show the distribution of DsRED2^+^ and EGFP^+^ cells and are overlaid with the raw data, where each dot represents an individual. (**D**) The mRNA level of il1b, tnfa, and cxcl8b.1 were determined by RT-qPCR and normalized against rps11 in 16 dpf larvae after treatment. The bars show the mean ± SEM of 20 pooled larvae (*n* = 20) and triplicate samples from 2 independent experiments (*n* = 2). ns, non-significant; **** *p* < 0.0001, according to Kruskal-Wallis followed by Dunn’s multiple comparison test (**A**), two-way ANOVA followed by Dunnett’s multiple comparison test (**C**), and Brown-Forsythe ANOVA followed by Dunnett’s T3 multiple comparison test (**D**). Scale bars, 500 μm.

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
