# Peer review of "Senescence-Independent Anti-Inflammatory Activity of the Senolytic Drugs Dasatinib, Navitoclax, and Venetoclax in Zebrafish Models of Chronic Inflammation"

_ijms, 2022, doi:10.3390/ijms231810468_

Round 1
Reviewer 1 Report
In the manuscript entitled “Senescence-independent anti-inflammatory activity of the senolytic drugs dasatinib, navitoclax and venetoclax in zebrafish models of chronic inflammation”, David Hernández-Silva and collaborators reported that senolytics drugs dasatinib, navitoclax and venetoclax have an anti-inflammatory effects, resulting in the amelioration of chronic inflammation.
In my opinion, this work has several critical issues listed below.
Inflammaging condition is defined as chronic pro-inflammatory status characterized by: i) high levels of proinflammatory markers; ii) chronic activation of innate immune system; iii) accumulation of senescence cells.
Spint1a-deficient zebrafish model is proposed in this work as a model of skin inflammaging condition in vivo
However, not b-gal staining was observed in the skin; furthermore the authors claim that “the Spint1a-deficient zebrafish line does not show inflammaging” (lines 99-100), even if they show a reduction of IL1b levels after treatment with senolytics. This is a bit confusing and should be explained better!!!
Always about Spint1a-deficient zebrafish model:
what is the meaning of b-gal positivity in the head and muscle of both wild type and Spint1a-deficient larvae?
What is the role of keratinocytes aggregates in the context of chronic inflammation (it is not so obvious for who is not expert in the field!)?
In addition to IL1b, some else pro-inflammatory markers of inflammaging should be evaluated.
About the High-Fat Diet zebrafish model:
Why were evaluated only navitoclax effects?
It is not explained as it is defined the inflammation scoring shown in figure 3C
Consistently with what was done for the other model, beta gal data should be shown, along with the evaluation of pro-inflammatory markers of inflammaging.
Author Response
Reviewer 1
In the manuscript entitled “Senescence-independent anti-inflammatory activity of the senolytic drugs dasatinib, navitoclax and venetoclax in zebrafish models of chronic inflammation”, David Hernández-Silva and collaborators reported that senolytics drugs dasatinib, navitoclax and venetoclax have an anti-inflammatory effects, resulting in the amelioration of chronic inflammation.
In my opinion, this work has several critical issues listed below.
Inflammaging condition is defined as chronic pro-inflammatory status characterized by: i) high levels of proinflammatory markers; ii) chronic activation of innate immune system; iii) accumulation of senescence cells.
Spint1a-deficient zebrafish model is proposed in this work as a model of skin inflammaging condition in vivo
However, not b-gal staining was observed in the skin; furthermore the authors claim that “the Spint1a-deficient zebrafish line does not show inflammaging” (lines 99-100), even if they show a reduction of IL1b levels after treatment with senolytics. This is a bit confusing and should be explained better!!!
We are sorry for this misunderstanding. Our initial aim was to use models of inflammaging but, unfortunately, none of the 2 models used show senescence. Therefore, we evaluated the anti-inflammatory effects of the senolytics, as indicated in the title and explained in the abstract. We think the results are of interest, since they show that the 3 drugs have anti-inflammatory effect independently of their senolytic activity.
Always about Spint1a-deficient zebrafish model:
what is the meaning of b-gal positivity in the head and muscle of both wild type and Spint1a-deficient larvae?
Senescence has been shown to be involved in development, so this staining confirms that our protocol is working and that skin inflammation does not lead to senescence.
What is the role of keratinocytes aggregates in the context of chronic inflammation (it is not so obvious for who is not expert in the field!)?
Keratinocyte aggregates results from the hyperproliferation characteristic of this model. So, they are used as surrogate of keratinocyte proliferation. This has been clarified in the Introduction.
In addition to IL1b, some else pro-inflammatory markers of inflammaging should be evaluated.
As indicated above, we did not find inflammaging in the Spint1a-deficient model. We have included the expression of tnfa and cxcl8b.1 genes and the results were confirmed.
About the High-Fat Diet zebrafish model:
Why were evaluated only navitoclax effects?
The experimental set up of this experiment is complicated, since fish must be fed with HCD for10 days. As it was not possible to perform this experiment with the 3 drugs, we decided to use Navitoclax that ameliorated all inflammatory markers in the Spint1a-deficient model.
It is not explained as it is defined the inflammation scoring shown in figure 3C
We are sorry. This is indeed a semi-quantification of the number of neutrophils and macrophages. As we have quantified the number of macrophages and neutrophils per larva, we have removed this data because they are redundant and confusing.
Consistently with what was done for the other model, beta gal data should be shown, along with the evaluation of pro-inflammatory markers of inflammaging.
We have analyzed the expression of il1b, tnfa and cxcl8b.1 and, surprisingly, navitoclax did not reduced their expression. Therefore, navitoclax is directly affecting emergency myelopoiesis. This novel and interesting observation was appropriately discussed.

Reviewer 2 Report
This paper describes a topic on the anti-inflammatory activity of several senolytic drugs in the context of chronic inflammation, by using two different zebrafish models: (i) a chronic skin inflammation model with a hypomorphic mutation in spint1a, the gene encoding the serine protease inhibitor, kunitz-type, 1a (also known as hai1a) and (ii) a non-alcoholic fatty liver disease/non-alcoholic steatohepatitis (NAFLD/NASH) model with inflammation induced by a high-fat diet. Our results show that, although these models do not manifest premature aging, the senolytic drugs dasatinib, navitoclax and venetoclax have an anti-inflammatory effect that results in the amelioration of chronic inflammation. Some key aspects are left out. Therefore, the paper needs mandatory revise before it can be accepted. Meanwhile, the following comments should be addressed before publications.
1. It is strongly recommended that the authors should mention clearly the newly developed and /or found point of in section introduction, compared with papers by introducing anti-inflammatory activity of several senolytic drugs in the context of chronic inflammation already reported in this field.
2. The authors present the results of anti-inflammatory activity of several senolytic drugs are currently widely well known in research area, but the technical and academic descriptions are still deficient. The authors should provide more technical and academic descriptions on what different/ effect compared with published results of anti-inflammatory activity of several senolytic drugs.
3. The authors should show the scale bar in Figures 1(A), 1(C); Figures 2(B), 2(C) ; Figures 3(C), 2(D).
4. The authors should compare clearly what the difference for the anti-inflammatory activity of several senolytic drugs, how to improve the anti-inflammatory activity, and how to define and change the anti-inflammatory activity?
5. The authors should discuss and compare what reliability for anti-inflammatory activity of several senolytic drugs design.
6. The authors should show and compare the quantitative data of reliability for anti-inflammatory activity of several senolytic drugs.
7. Adding the references including 2020-2022 is recommended.
Author Response
Reviewer 2:
This paper describes a topic on the anti-inflammatory activity of several senolytic drugs in the context of chronic inflammation, by using two different zebrafish models: (i) a chronic skin inflammation model with a hypomorphic mutation in spint1a, the gene encoding the serine protease inhibitor, kunitz-type, 1a (also known as hai1a) and (ii) a non-alcoholic fatty liver disease/non-alcoholic steatohepatitis (NAFLD/NASH) model with inflammation induced by a high-fat diet. Our results show that, although these models do not manifest premature aging, the senolytic drugs dasatinib, navitoclax and venetoclax have an anti-inflammatory effect that results in the amelioration of chronic inflammation. Some key aspects are left out. Therefore, the paper needs mandatory revise before it can be accepted. Meanwhile, the following comments should be addressed before publications.
- It is strongly recommended that the authors should mention clearly the newly developed and /or found point of in section introduction, compared with papers by introducing anti-inflammatory activity of several senolytic drugs in the context of chronic inflammation already reported in this field.
- The authors present the results of anti-inflammatory activity of several senolytic drugs are currently widely well known in research area,but the technical and academic descriptions are still deficient. The authors should provide more technical and academic descriptions on what different/ effect compared with published results of anti-inflammatory activity of several senolytic drugs.
1-2:The effects of senolytic drugs are commented in the 4th paragraph of the Introduction. We do not find appropriate to discuss in the Introduction previous data with our results.
- The authors should show the scale bar in Figures 1(A), 1(C); Figures 2(B), 2(C) ; Figures 3(C), 2(D).
The bars have been included.
- The authors should compare clearly what the difference for the anti-inflammatory activity of several senolytic drugs, how to improve the anti-inflammatory activity, and how to define and change the anti-inflammatory activity?
- The authors should discuss and compare what reliability for anti-inflammatory activity of several senolytic drugs design.
- The authors should show and compare the quantitative data of reliability for anti-inflammatory activity of several senolytic drugs.
4-6:The anti-inflammatory activity of senolytic is discussed in the last paragraph of the Discussion. Unfortunately, there is no information about the anti-inflammatory effects of the senolytic drugs used in our study.
- Adding the references including 2020-2022 is recommended.
We have checked the references and found them appropriate.

Round 2
Reviewer 2 Report
The anti-inflammatory effects of the senolytic drugs used in author's work is still lacking.
Author Response
As it has not been rerported anti-inflammatory activities of dasatinib, navitoclax and venetoclax independent of their senolytic activity, we have clarified this fact in the discussion of the revised version.